# Influences of Ni Content on the Microstructural and Catalytic Properties of Perovskite LaNi$_x$Cr$_{1-x}$O$_3$ for Dry Reforming of Methane

Tingting Zhao, Haoran Yu, Xuyingnan Tao, Feiyang Yu, Ming Li and Haiqian Wang *

Hefei National Research Center for Physical Science at the Microscale, University of Science and Technology of China, Hefei 230026, China
* Correspondence: hqwang@ustc.edu.cn

**Abstract:** Perovskite oxides were widely used as precursors for developing metal-support type catalysts. It is attractive to explore the catalytic properties of the oxides themselves for dry reforming of methane (DRM). We synthesized LaNi$_x$Cr$_{1-x}$O$_3$ (x = 0.05–0.5) samples in powder form using the sol-gel self-combustion method. Ni atoms are successfully doped into the LaCrO$_3$ perovskite lattice. The perovskite grains are polycrystalline, and the crystallite size decreases with increasing Ni content. We demonstrated that the LaNi$_x$Cr$_{1-x}$O$_3$ perovskites show intrinsically catalytic activity for DRM reactions. Reducing the Ni content is helpful to reduce carbon deposition resulting from the metal Ni nanoparticles that usually coexist with the highly active perovskite oxides. The CH$_4$ conversion over the LaNi$_{0.1}$Cr$_{0.9}$O$_3$ sample reaches approximately 84% at 750 °C, and the carbon deposition is negligible.

**Keywords:** heterogeneous catalysis; perovskite phases; intrinsic activity; dry reforming of methane; coke resistance

## 1. Introduction

Dry reforming of methane (DRM) is an important reaction that converts two greenhouse gases, CH$_4$ and CO$_2$, to valuable syngas, H$_2$ and CO, with an H$_2$/CO ratio close to 1, which is suitable for synthesizing long-chain hydrocarbon chemicals through the Fischer-Tropsch reaction [1,2]. DRM is also attractive in saving the cost of CO$_2$ separation when CO$_2$-rich CH$_4$ gas, such as biogas, is used as a feedstock to produce syngas [3,4]. Catalysts play a crucial role in a DRM reaction because both CH$_4$ and CO$_2$ are very stable molecules, and the reaction kinetics at economic reaction temperatures will be very sluggish without a high-performance catalyst. A DRM reaction is endothermic. To achieve acceptable CH$_4$ and CO$_2$ conversions limited by thermodynamics, the reaction needs to be carried out at temperatures typically higher than 600 °C [5]. Side reactions, such as the CH$_4$ decomposition reaction (CH$_4$ = 2H$_2$ + C) and the Boudouard reaction (2CO = CO$_2$ + C), may result in coking that will deactivate the catalysts or even block the reactor [6,7]. The reverse water–gas shift side reaction (RWGS, CO$_2$ + H$_2$ = CO + H$_2$O) always occurs simultaneously with DRM, which reduces the H$_2$/CO ratio to lower than 1. Thus, an ideal catalyst for DRM should be highly active, coking resistant, and thermostable [3].

Ni is the most investigated transition metal element for a DRM reaction because of its high catalytic activity and low cost compared to noble metals. However, supported Ni nanoparticles that act as catalytic active centers in metal-support catalysts suffer from coking and sintering problems [8–10]. Perovskite oxides were widely used as precursors for developing metal-support type catalysts because the in situ formation of highly dispersed Ni nanoparticles on oxide support may improve the activity and suppress coking [11,12]. What is more, the feasible application of doping of noble or non-noble metal atoms in perovskite precursors can improve the catalysts by further reducing the size of active

metal nanoparticles, introducing active oxygen species in the support, and tailoring the metal-support interactions [13–17]. It has been reported that small Ni nanoparticles have strong anti-coking resistance [18]. An extreme case is that the single-atom catalyst with isolated Ni atoms dispersed over hydroxyapatite (HAP) is highly active and completely coke-resistant during high-temperature DRM [18].

Other than the usual metal-support type catalysts and single-atom catalysts, it is reported that perovskite oxides with the general formula $ABO_3$ (A: lanthanide or alkaline earth metal; B: transition metal) show intrinsic activity for many reactions [19–21]. Perovskites have good flexibility and diversity in their chemical composition and can accommodate solid defects, such as vacancies at both the cation and anion sites [22,23]. The B-site transition metals on the surface are believed to be the active centers owing to the exposed d electron orbitals (e.g., Ni 3d). Considering that the B-site transition metals embedded in the perovskite lattice are atomically dispersed, we can expect to develop highly active and anti-coking catalysts for DRM. For instance, Ni-containing perovskites have been developed and proved to be highly active [22,24]. However, the usual Ni-containing perovskites, such as $LaNiO_3$ and $La(NiFe)O_3$, are not stable under DRM conditions and will be over-reduced to metal-support catalysts [8,25,26]. On the other hand, perovskite $LaCrO_3$ is very stable in both reducing and oxidizing environments, but it is catalytically inert for DRM [27,28]. We recently demonstrated that Ni-containing perovskite oxides in the two-dimensional submonolayer (SML) form, such as $LaNiO_\Delta$-SML and $La(NiCo)O_\Delta$-SML, can be stabilized by a perovskite $LaCrO_3$ support and used for catalyzing a DRM reaction [29]. The interesting point is that Ni atoms in the low-valent oxide form are highly active and anti-coking for a DRM reaction, even though the long-term stability of $LaNiO_\Delta$-SML needs to be further improved. Understanding the microstructural and catalytic properties of such materials will pave an attractive way for us to explore atomically dispersed catalysts.

In the present work, perovskite $LaNi_xCr_{1-x}O_3$ ($x \leq 0.5$) catalysts were synthesized and characterized. The influences of the Ni content on the microstructural and catalytic properties for catalyzing the DRM reaction are discussed. The intrinsic activity of perovskite oxides is confirmed.

## 2. Results and Discussion

### 2.1. Crystalline Structure and Specific Surface Areas

Figure 1a shows that the XRD patterns of all the fresh $LaNi_xCr_{1-x}O_3$ (x = 0.05–0.5) samples are dominated by a well-defined $ABO_3$ perovskite phase with the space group of Pbnm ($LaCrO_3$: JCPDS 00-71-1231). The peaks at approximate 22.9°, 32.6°, 40.1°, 46.7°, 52.6°, 58.1°, and 68.4° correspond to (002), (112), (022), (004), (222), (132), and (224) planes, respectively. The NiO phase appears as x increases to above 0.2, and the intensity of the NiO peak increases with x, indicating that the solubility of Ni cations in the perovskite is limited (see the right panel of Figure 1a). However, no $La_2CrO_6$, $La_2O_3$, or its derivatives can be detected, indicating that the perovskite is B-site deficient or that the $La_2O_3$ or its derivatives exist in an amorphous form. As x increases, the XRD diffraction peaks of the perovskite slightly shift to higher 2θ angles. For example, the (022) peak belonging to the perovskite becomes broadened and shifts slightly from 40.3 to 40.7° as x increases from 0.1 to 0.5 (see the right panel of Figure 1a), suggesting that the perovskite lattice shrinks as more Cr is replaced by Ni because the standard six-coordinate ionic radius of $Ni^{3+}$ (0.60 Å) is smaller than that of $Cr^{3+}$ (0.615 Å). This phenomenon agrees with that reported by Yang [30].

To understand the microstructural evolution of the catalysts induced by the $H_2$ activation and DRM reaction, we examined the XRD patterns of the reduced and used catalysts. As shown in Figure 1b,c, the main features of the catalyst remain the same as those of the fresh catalyst. The perovskite phase dominates the XRD, and no $La_2O_3$ or its derivatives can be detected, indicating that the $LaCrO_3$ perovskite structure is very stable. The Ni phase observed in the reduced and used samples with $x \geq 0.2$ (shown in the right panel of Figure 1b,c, magnified by 20 times) comes from the reduction of NiO, as well as the Ni atoms exsolved from the $LaNi_xCr_{1-x}O_3$ perovskite. No Ni phase can be observed in

the samples with x ≤ 0.1, which should be because of the very low Ni content or the Ni atoms being highly dispersed. The increased intensity in the XRD diffraction peak at approximately 26.3° for the x ≥ 0.2 used samples indicates that heavy carbon deposition occurred in the catalysts when the Ni loading was high (see the right panel of Figure 1c).

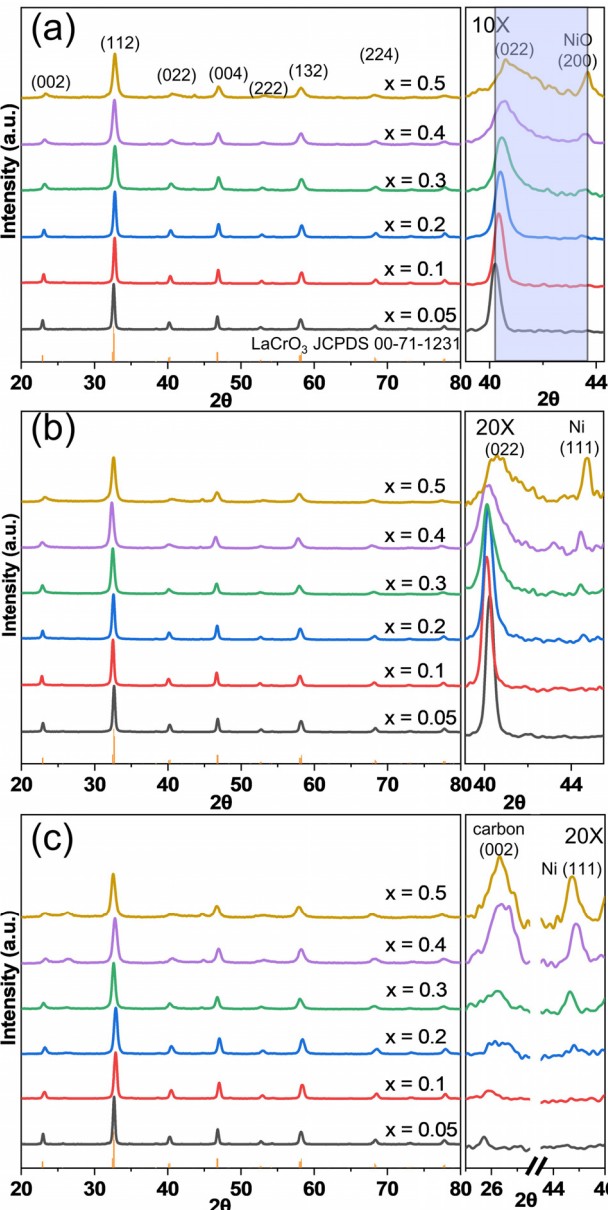

**Figure 1.** XRD patterns of fresh (**a**), reduced (**b**), and used (**c**) $LaNi_xCr_{1-x}O_3$ samples. The time on stream of used samples is 10 h, except for the x = 0.5 sample, which has a 4.5 h time on stream due to heavy carbon deposition. The lattice planes labeled in the figure are those of the perovskite phase except otherwise specified. Perovskite: orthorhombic, Pbnm, JCPDS 00-71-1231. NiO: Cubic, Fm-3m, JCPDS 00-89-7130. Ni: Cubic, Fm-3m, JCPDS 00-87-0712.

Table 1 shows the average crystallite sizes of the perovskite in the fresh, reduced, and used samples calculated by refining the (002), (112), (022), (004), and (132) XRD peaks with MDI Jade software. The average crystallite size of the fresh sample decreases with increasing Ni content. This is because the periodicity of the $LaCrO_3$ perovskite lattice is disturbed by the doping of Ni atoms. The defects produced around the dopant may create charge imbalance and oxygen vacancies and introduce lattice strain, which in turn increase the amorphous nature leading to the decrease in the crystallite size [31]. As an increasing

number of Ni ions are doped into the perovskite lattice, the increased concentration of solid defects impedes the grains from growing larger [32]. The same changing trend of the crystallite size with Ni content was also observed in the reduced and used samples. It is interesting to note that the average crystallite sizes of the used samples are smaller than the corresponding fresh ones even though they were sintered under DRM conditions at 750 °C for many hours. This should be related to the migration and aggregation of the point defects (zero dimension, 0D), such as Ni ions and O vacancies. These point defects may aggregate into larger ones, leaving behind aggregated defects inside the perovskite grains, such as dislocations (1D), grain boundaries (2D), or even microcracks, which make the crystalline grains smaller. It is also possible that Ni ions and O vacancies migrate from the inside of a $LaNi_xCr_{1-x}O_3$ crystalline grain out to the surface or interface, making the perovskite grain smaller and denser.

**Table 1.** The average crystallite size of fresh, reduced, and used $LaNi_xCr_{1-x}O_3$ samples.

| x in $LaNi_xCr_{1-x}O_3$ | Average Crystallite Size/nm | | |
|:---:|:---:|:---:|:---:|
| | **Fresh** | **Reduced** | **Used** |
| 0.05 | 29.1 | 26.1 | 28.4 |
| 0.1 | 25.8 | 25.8 | 19.3 |
| 0.2 | 21.1 | 21.8 | 16.0 |
| 0.3 | 15.1 | 17.3 | 16.3 |
| 0.4 | 13.7 | 14.9 | 12.4 |
| 0.5 | 12.7 | 13.9 | 12.0 |

Note: The average crystallite sizes in Table 1 were determined from the (002), (112), (022), (004), and (132) XRD peaks.

Different from the changing trend of the perovskite crystallite size determined by XRD, the BET-specific surface area of fresh $LaNi_xCr_{1-x}O_3$ samples is between 7–10 $m^2\ g^{-1}$, and it did not show an increasing trend with the Ni content (see Table 2), indicating that the perovskite grains are polycrystalline, so the BET surface area did not change much. The relatively smaller BET-specific surface area is common for perovskite powders prepared by the sol–gel combustion method because the synthesis process temperature is high [33].

**Table 2.** The BET-specific surface area of fresh $LaNi_xCr_{1-x}O_3$ samples.

| x in $LaNi_xCr_{1-x}O_3$ | Specific Surface Area ($m^2\ g^{-1}$) | Correlation Coefficient |
|:---:|:---:|:---:|
| 0.05 | 9.7 | 0.9999 |
| 0.1 | 9.4 | 0.9977 |
| 0.2 | 7.2 | 0.9984 |
| 0.3 | 9.4 | 0.9996 |
| 0.4 | 9.9 | 0.9997 |
| 0.5 | 9.5 | 0.9998 |

*2.2. Microstructure*

Figures 2 and 3 compare some typical HR-TEM images and relative EDS mapping of Ni in the fresh and used $LaNi_xCr_{1-x}O_3$ samples for x = 0.1, 0.3, and 0.5. Both the fresh and used samples mainly consist of perovskite grains, which can be confirmed by matching the spacing between the fringes in the HR-TEM images to the d-spacing determined by XRD. Solid defects, including grain boundaries, can be found, especially in the samples with high Ni content, confirming that the perovskite grains are polycrystalline (see Figures 2c and 3c). The EDS mapping images of the fresh samples (Figure 2 and Figures S1, S3, S5, and S6 in Supplementary Materials) show that La, Cr, Ni, and O elements are distributed homogeneously in the perovskite grains. Nevertheless, NiO grains can be observed in the fresh samples with high Ni contents (x = 0.3 and 0.5). These NiO grains are recognized in the regions with a brighter Ni signal and weaker Cr and La signals in the

EDS mapping images, for example, see Figure S5 (x = 0.5). This observation agrees with our XRD analyses.

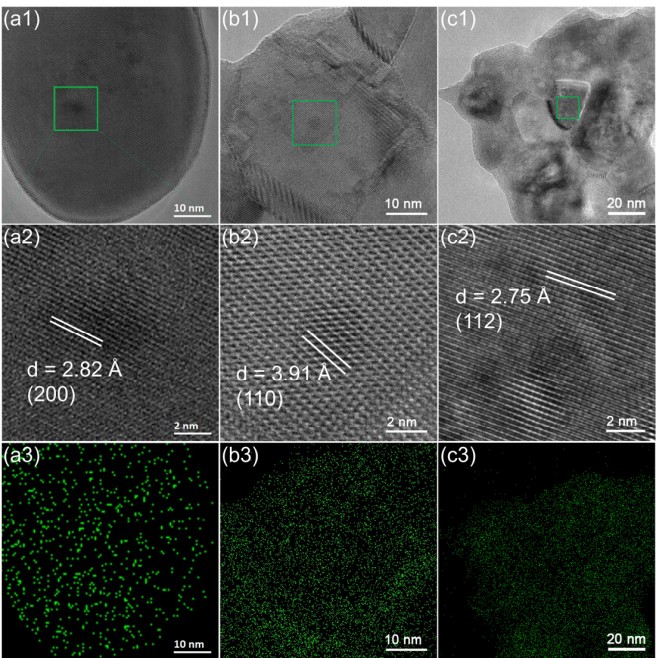

**Figure 2.** TEM and EDS mapping images of fresh LaNi$_x$Cr$_{1-x}$O$_3$ samples with x = 0.1 (**a1–a3**), 0.3 (**b1–b3**), and 0.5 (**c1–c3**). The first row shows TEM images, and the second row shows the corresponding magnified images of green squares in the first row. The third row shows EDS mapping images of Ni elements in the first row. The d-spacing values and their corresponding lattice planes of the perovskite oxides are labeled in the magnified TEM images.

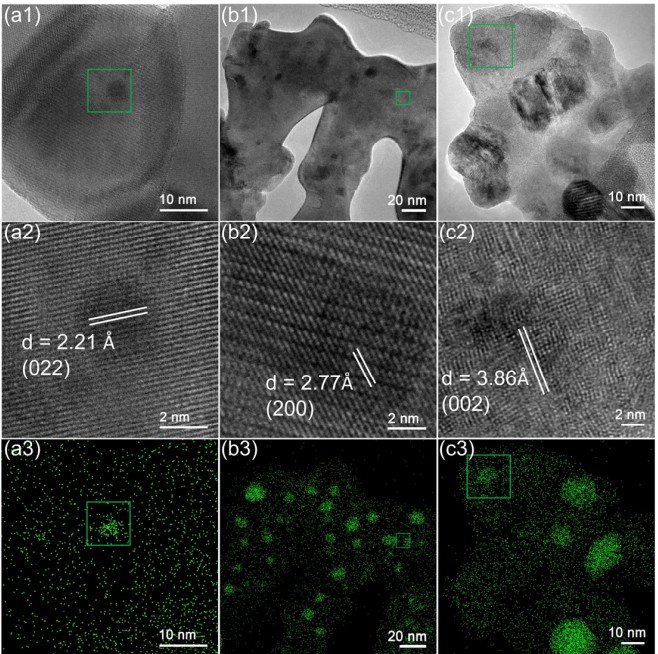

**Figure 3.** TEM and EDS mapping images of the used LaNi$_x$Cr$_{1-x}$O$_3$ samples with x = 0.1 (**a1–a3**), 0.3 (**b1–b3**), and 0.5 (**c1–c3**). The first row shows TEM images, and the second row shows the corresponding magnified images of green squares in the first row. The third row shows EDS mapping images of Ni elements in the first row. The d-spacing values and their corresponding lattice planes of the perovskite oxides are labeled in the magnified TEM images.

After the catalysts were used for DRM reactions, we can see many Ni-rich regions from the EDS mapping corresponding to the dark regions in the HR-TEM images (Figure 3b,c). The Ni-rich regions share the same lattice with the perovskite grains. It has been proven in our recent work that these regions are two-dimensional (2D) perovskite oxide, $LaNiO_\Delta$ submonolayers (SML), which are highly active for DRM reactions [29]. The surface coverage of SMLs increases with x in the $LaNi_xCr_{1-x}O_3$ samples. In addition to SMLs on the surface of perovskite grains, we can also see many Ni metal particles and filamentous carbon in the samples with high Ni content (Figures S7 and S8, x = 0.5). Some of the Ni nanoparticles are detached from the perovskite grains by filamentous carbon, indicating that the filamentous carbon is very likely induced by the Ni nanoparticles and that the interaction between the metal Ni nanoparticles and the perovskite support is weak. No deposited carbon can be observed in the SML regions, suggesting that SMLs have a good anti-coking ability. Considering that the Ni ions in the SML are embedded in the perovskite-like oxide, we think the atomic-scale dispersion of Ni ions should help suppress the nucleation and growth of filamentous carbon, which is harmful to a DRM reaction.

### 2.3. Electronic Structure

Ni is the active element in the $LaNi_xCr_{1-x}O_3$ catalysts. The surface distribution of Ni and its interaction with neighboring atoms can be characterized by the surface-sensitive XPS technique. We use Ni 3p spectra for the characterization because the stronger Ni 2p XPS spectra overlap heavily with La 3d [34]. Figure 4a shows the Cr 3s and Ni 3p XPS spectra of fresh and used samples. As expected, the surface Ni content increases with x in both the fresh and used samples. It is noted that the surface Ni contents of the used samples are less than those of the corresponding fresh samples, indicating that some of the Ni atoms aggregated into large particles and that the lateral size of the metal Ni nanoparticles is larger than the probing depth (2–5 nm) of XPS [29].

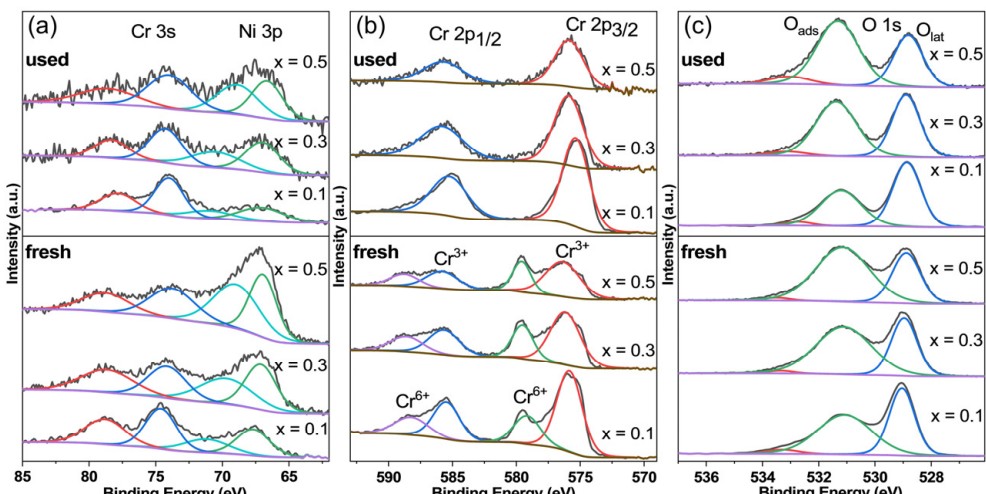

**Figure 4.** XPS spectra of the $LaNi_xCr_{1-x}O_3$ sample with x = 0.1, 0.3, and 0.5. (**a**) Cr 3s and Ni 3p spectra. (**b**) Cr 2p spectra. (**c**) O 1s spectra. The top half of Figure 4 shows the used samples, and the bottom half shows the fresh samples.

The energy separation (ΔE) (Table 3) of the Cr 3s multiplet splitting spectra depends on the charge transfer between Cr and Ni atoms in the $LaNi_xCr_{1-x}O_3$ perovskite and has a positive correlation with the Ni content in B-sites of $LaNi_xCr_{1-x}O_3$ [29]. ΔE increases with x in the fresh samples, indicating that the Ni content in the perovskite phase increases, and some of the Ni atoms may even exist in the NiO phase, as evidenced by the above XRD analyses. The ΔE of the used samples is smaller than that of the fresh samples, indicating that some of the Ni atoms are exsolved from the perovskite phase.

**Table 3.** $\Delta E$ of Cr 3s for fresh and used $LaNi_xCr_{1-x}O_3$ samples in Cr 3s spectra.

| x in $LaNi_xCr_{1-x}O_3$ | $\Delta E$ of Cr 3s for Fresh Samples (eV) | $\Delta E$ of Cr 3s for Used Samples (eV) |
|---|---|---|
| 0.1 | 4.2 | 3.8 |
| 0.3 | 4.5 | 4.0 |
| 0.5 | 5.0 | 4.6 |

Figure 4b shows the Cr 2p XPS spectra of fresh and used samples of x = 0.1, 0.3 and 0.5. The strong spin–orbit interaction splits the Cr 2p main peaks into Cr $2p_{3/2}$ and Cr $2p_{1/2}$ doublets separated by ~10 eV. The Cr 2p spectra of the fresh sample show two sets of Cr 2p doublets that belong to $Cr^{3+}$ and $Cr^{6+}$, respectively. The peaks located at approximately 576 eV (Cr $2p_{3/2}$) and 585 eV (Cr $2p_{1/2}$) can be ascribed to $Cr^{3+}$, while the peaks located at approximately 580 and 589 eV arise from $Cr^{6+}$ [35,36]. Although the $Cr^{6+}$ XPS peaks in the fresh samples are prominent, especially in the samples with high Ni content, we did not detect any $Cr^{6+}$ compounds in the XRD (see Figure 1). This is because $La_2CrO_6$ mainly exists on the surface of the perovskite, which can easily be detected by XPS [37].

$Cr^{6+}$ increases with Ni content in the fresh catalysts should be related to the charge disproportion effect in the perovskite [38]. In the fresh catalysts, $Ni^{3+}$ and $Cr^{3+}$ occupy the B-site of perovskite $LaNi_xCr_{1-x}O_3$ and are stabilized by the $BO_6$ octahedral crystal field. However, $Ni^{3+}$ tends to become $Ni^{2+}$ because $Ni^{2+}$ is a stable oxidation state of Ni. The reduced Ni oxidation state will in turn drive the $Cr^{3+}$ to $Cr^{6+}$ to keep charge neutrality in the perovskite. This charge disproportion effect is more likely to happen at the surface region because the surface lattice relaxation weakens the crystal field, and thus reduces the stability of $Ni^{3+}$ and $Cr^{3+}$. This explains why the $Cr^{6+}$ is sensitive to XPS but not XRD, as well as why $Cr^{6+}$ increases with Ni content in the fresh catalysts.

After the DRM test, most of the Ni atoms are exsolved from the perovskite lattice out to the surface, and thus the charge disproportion effect between Ni and Cr no longer dominates the oxidation state of Cr. What is more, the DRM atmosphere is a relatively reducing atmosphere (with $H_2$ and CO in the product). Thus, no $Cr^{6+}$ can be observed by XPS in the used catalysts because it is reduced to $Cr^{3+}$.

Figure 4c shows the O 2p XPS spectra of the fresh and used $LaNi_xCr_{1-x}O_3$ samples with x = 0.1, 0.3 and 0.5. The peak around 529 eV, denoted as $O_{lat}$, can be assigned to lattice oxygen ($O^{2-}$) in the perovskite and NiO oxides, while the peak around 531 eV, denoted as $O_{ads}$, comes from surface adsorption oxygen species and other hydroxyls (OH) and carbonate species ($CO_3^{2-}$), whose intensity reflects the concentration of oxygen vacancy in the perovskite [39,40]. The surface oxygen species usually relate to defects/oxygen vacancies since they can act as absorption centers [41]. The peak at about 533 eV is usually considered to be correlated to adsorbed molecular water [42]. The area ratio of $O_{ads}/O_{lat}$ (Table S1) in the XPS spectra increases with the Ni content in the fresh samples, suggesting that there are more oxygen vacancies in the samples with high Ni content. Or in other words, $\delta$ in $LaNi_xCr_{1-x}O_{3-\delta}$ increases with x, indicating that the perovskite becomes more oxygen deficient as more Ni atoms are doped. The used samples show a smaller $O_{ads}/O_{lat}$ ratio as compared to the fresh ones, indicating that some oxygen vacancies migrated out of the perovskite with the exsolvement of Ni atoms. It is reported the active oxygen species related to the oxygen vacancies help reduce carbon deposition [43]. However, as we see in Figure 4c and Table S1 that the $O_{ads}/O_{lat}$ ratio in the used samples increases with Ni content, while we know from the XRD, TEM, and TPO (will be discussed later) analyses that carbon deposition is more severe in the high Ni content samples. Thus, the increases in $O_{ads}/O_{lat}$ ratio cannot compensate for the increased carbon deposition trend that resulted from the increase in Ni loading.

### 2.4. Reducibility

Figure 5 shows the TG and DTG profiles of fresh $LaNi_xCr_{1-x}O_3$ samples obtained by $H_2$-TPR measurements. The TG profiles show that the samples are reduced in two steps.

The first step is between 200 and 360 °C, and the second step is between 370 and 570 °C. Correspondingly, two well-resolved peaks are observed in each of the DTG profiles. Based on the above XRD analysis and literature survey [44,45] we can assign the low-temperature reduction step to the reduction of $Ni^{3+}$ to $Ni^{2+}$ in the $LaNi_xCr_{1-x}O_3$ perovskite and the high-temperature step to the reduction of NiO to Ni. Thus, we see that both the Ni content in the perovskite and the amount of NiO increase with x in the $LaNi_xCr_{1-x}O_3$ samples, and the Ni content in the perovskite is higher than that in NiO. It is also noted that no clear reduction step corresponding to the reduction of $Ni^{2+}$ to $Ni^0$ in the $LaNi_xCr_{1-x}O_3$ perovskite can be resolved up to 900 °C in the TPR profiles, indicating that $Ni^{2+}$ is rather stable in the perovskite and difficult to reduce to $Ni^0$. Stojanović et al. [45] reported that $LaNi_xCr_{1-x}O_3$ compounds with x < 0.5 did not reduce to nickel metal in an $H_2$ atmosphere at <900 °C. Nevertheless, the slowly decreasing trend in the TG profiles in the high-temperature section suggests that at least some of the $Ni^{2+}$ cations in the perovskite were gradually reduced to $Ni^0$ at high temperatures.

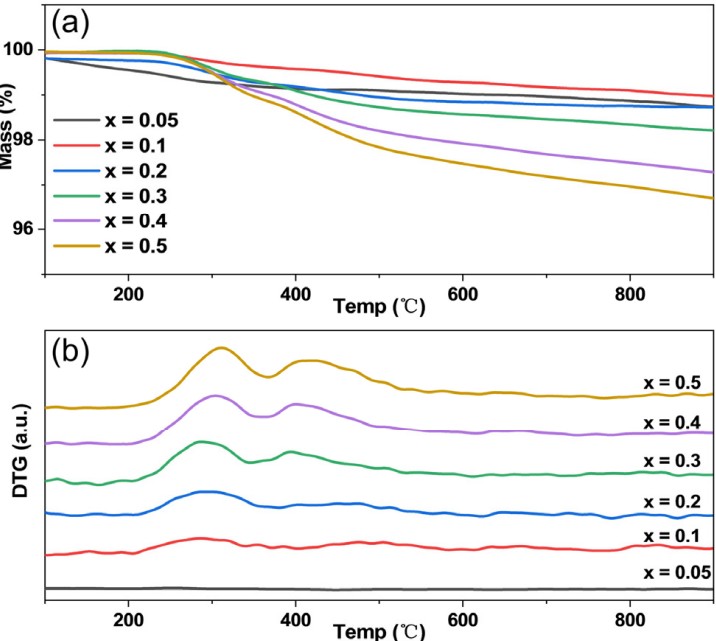

**Figure 5.** $H_2$-TPR profiles of fresh $LaNi_xCr_{1-x}O_3$ samples. (**a**) Thermogravimetric curve (TG). (**b**) First-order derivative on the TG curve.

### 2.5. Catalytic Performance

Figure 6a,b shows that for the x = 0.1–0.3 samples, the $CH_4$ and $CO_2$ conversions are stable within the 10 h on stream test. When the Ni loading is very low (x = 0.05), the $CH_4$ and $CO_2$ conversions show a decreasing trend. Nevertheless, the very high initial activity of the x = 0.05 samples indicates that the dispersion of Ni atoms on the catalyst surface is very high. The fast drop in the activity of the x = 0.05 sample should be because the Ni-support interaction is not strong enough to prevent the migration and aggregation of the highly dispersed Ni atoms [29]. Similar quick deactivation behavior in very low Ni loading catalyst was also observed in $Ni/CeO_2$ [46]. On the other hand, when the Ni loading is too high (x = 0.4 and 0.5), the DRM reaction fails to proceed long because of the blockage of the fixed bed reactor by carbon deposition. The $CH_4$ conversions over the x = 0.4 and 0.5 samples increase sharply at the end of the test, indicating that the $CH_4$ decomposition side reaction dominates the carbon deposition side reactions. The $H_2/CO$ ratio, $H_2$ selectivity, and carbon balance are also shown in Figure 6c–e, which will be discussed later. The mass loss in the TPO profiles (Figure 6f) between approximate 500–700 °C reflects the amount of deposited carbon, which increases remarkably with Ni content.

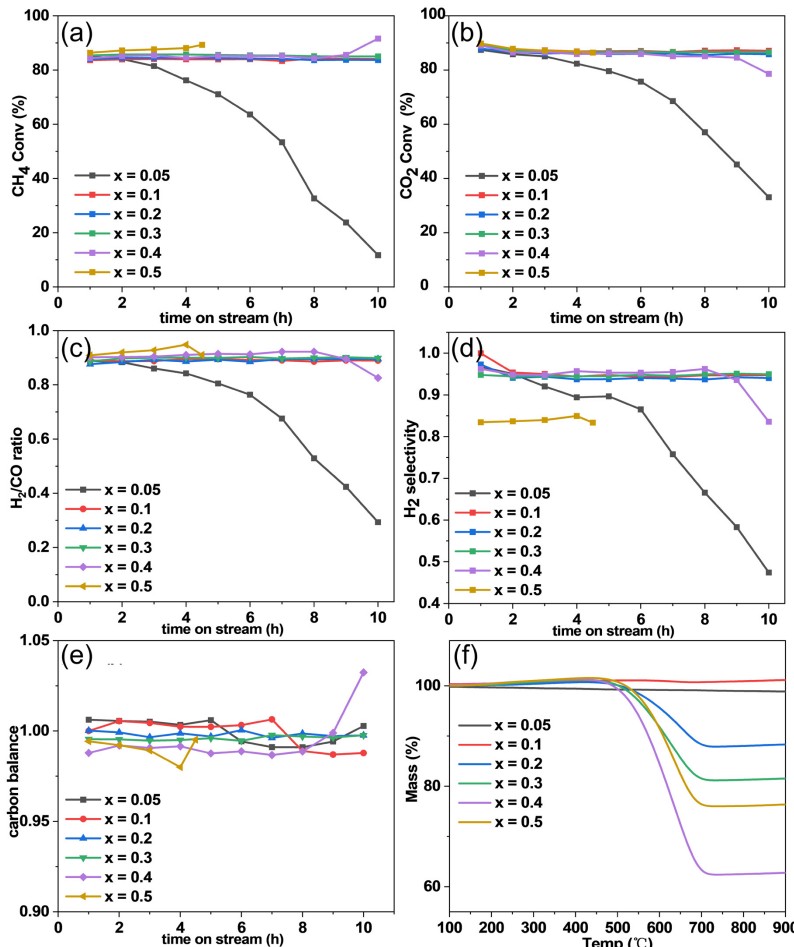

**Figure 6.** DRM performance of $LaNi_xCr_{1-x}O_3$ under different conditions. $CH_4$ conversion (**a**), $CO_2$ conversion (**b**), $H_2/CO$ ratio (**c**), $H_2$ selectivity (**d**), and carbon balance (**e**) as a function of time on stream during the DRM reactions over $LaNi_xCr_{1-x}O_3$ with different x values at 750 °C. (**f**) TPO profiles of used $LaNi_xCr_{1-x}O_3$ samples.

To investigate the influence of Ni loading on the catalytic performance, we compared the $CH_4$ and $CO_2$ conversions, $H_2/CO$ ratio, $H_2$ selectivity, and carbon balance of the x = 0.05–0.5 catalysts at 2 h on stream (Figure 7). The $CH_4$ conversion slightly increases from 83% to 87% as the Ni loading increases from x = 0.05 to 0.5, implying that a higher Ni loading is favorable for the main DRM reaction. The $CO_2$ conversion shows the same changing trend as that of $CH_4$ conversion but is a little higher than the $CH_4$ conversion owing to the RWGS side reaction. Meanwhile, the $H_2/CO$ ratio also increases from 89% to 92%. The $H_2/CO$ ratio for all the samples is smaller than 1 because of the RWGS side reaction ($H_2 + CO_2 = CO + H_2O$), which consumes $H_2$ and generates an extra amount of CO. The $H_2$ selectivity, which depends on the $H_2$ supply from the converted $CH_4$ and the amount of $H_2$ consumed by the RWGS reaction, is around 95% for all the catalysts. The carbon balance is very close to 1 but shows a slightly decreasing trend with increasing Ni content due to carbon deposition. The carbon deposition rate of the catalysts can be more precisely determined from the TPO results (see Figure 6f) and is also illustrated in Figure 7. The carbon deposition rate shows a monotonous and quick increase from 0.02 $mg_C$ $g_{cat}^{-1}$ $h^{-1}$ (x = 0.05) to 76.2 $mg_C$ $g_{cat}^{-1}$ $h^{-1}$ (x = 0.5) with increasing Ni loading. Thus, the x = 0.1 sample is preferred because it has a relatively low Ni loading (2.5 wt%) and shows high catalytic activity and negligible carbon deposition.

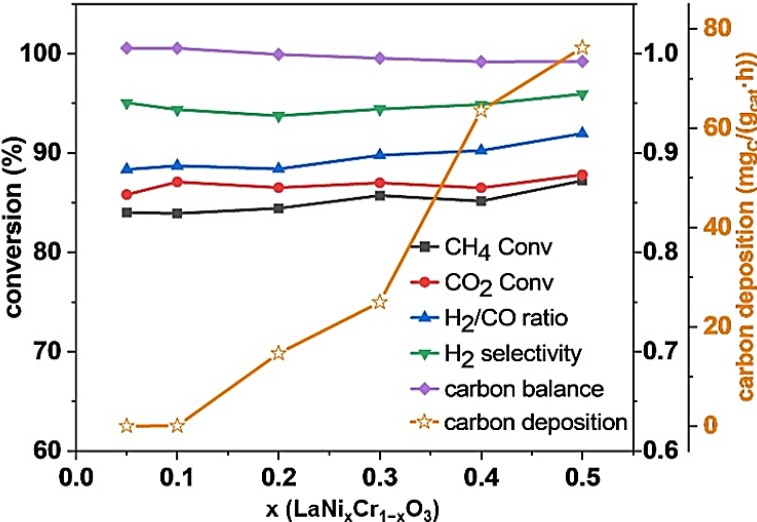

**Figure 7.** Changes in $CH_4$ conversion, $CO_2$ conversion, $H_2/CO$ ratio, $H_2$ selectivity, carbon balance and carbon deposition with x in $LaNi_xCr_{1-x}O_3$ at 750 °C. Condition: 12 L g$^{-1}$ h$^{-1}$, $CH_4$:$CO_2$ = 1. The data reported in Figure 7 are those collected at a reaction time of 2 h.

The temperature-dependent activity of the x = 0.1 sample is shown in Figure 8. The $CH_4$ and $CO_2$ conversions are very close to the thermoequilibrium values from 600–850 °C, indicating that the catalyst is highly active. As expected, the $H_2/CO$ ratio and $H_2$ selectivity increase with temperature because a high temperature favors the main DRM reaction. The carbon balance is very close to 1 indicating that the x = 0.1 sample has good anti-coking properties. Moreover, we also tested the DRM performance of the x = 0.1 sample without $H_2$ activation (Figure 9). It is interesting to see that the sample shows an equivalent catalytic activity in terms of $CH_4$ and $CO_2$ conversions, $H_2/CO$ ratio, $H_2$ selectivity, and carbon balance compared to its counterpart with $H_2$ activation. The close $CH_4$ and $CO_2$ conversions indicate that the DRM main reaction dominates the total reaction while the RWGS side reaction is minor, as evidenced by the relatively high $H_2/CO$ ratio. The self-activation behavior of the sample implies that the $LaNi_xCr_{1-x}O_3$ perovskite oxides are highly active for a DRM reaction before the formation of SMLs and metal Ni nanoparticles. This should be because the perovskites are oxygen deficient, especially on the surface. Oxygen vacancies will drive the Ni cations embedded in the perovskite into their lower oxidation states, making the Ni 3d electron orbitals open to the reactants.

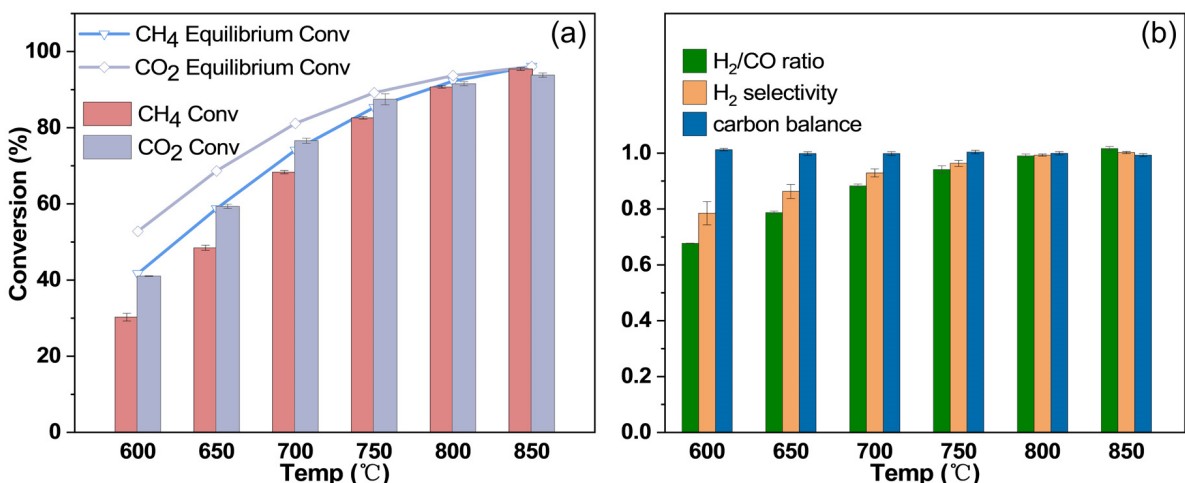

**Figure 8.** Temperature-dependent catalytic performance of the as-reduced $LaNi_{0.1}Cr_{0.9}O_3$ catalyst between 600–850 °C. Condition: 12 L g$^{-1}$ h$^{-1}$, $CH_4$:$CO_2$ = 1.

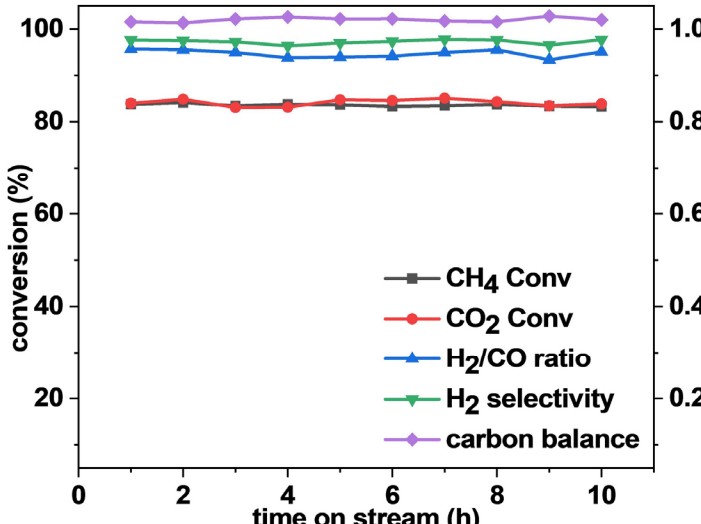

**Figure 9.** DRM catalytic performance of $LaNi_{0.1}Cr_{0.9}O_3$ without $H_2$ activation. Condition: 750 °C, 12 L g$^{-1}$ h$^{-1}$, $CH_4$:$CO_2$ = 1.

We compared our catalysts with those reported in the literature applied to DRM under similar conditions (Table 4). The optimized $LaNi_{0.1}Cr_{0.9}O_3$ with a low Ni loading performed well among the reported catalysts.

**Table 4.** DRM catalytic performance, as reported in the literature.

| Catalyst | GHSV | Temperature (°C) | CH$_4$ Conv (%) | CO$_2$ Conv (%) | Ref. |
|---|---|---|---|---|---|
| $LaNiO_3$ | 15 L g$^{-1}$ h$^{-1}$ | 750 | 99 | 95 | [11] |
| $LaNi_{0.8}Mn_{0.2}O_3$ | 15 L g$^{-1}$ h$^{-1}$ | 750 | 97 | 95 | [11] |
| $LaNi_{0.4}Ce_{0.6}O_3$ | 12 L g$^{-1}$ h$^{-1}$ | 800 | 93 | 93 | [12] |
| $La_{0.6}Ce_{0.4}Ni_{0.5}Fe_{0.5}O_3$ | 12 L g$^{-1}$ h$^{-1}$ | 750 | 62 | 72 | [14] |
| $La_{0.6}Ce_{0.4}Ni_{0.9}Zr_{0.01}Y_{0.09}O_3$ | 42 L g$^{-1}$ h$^{-1}$ | 800 | 89 | 91 | [16] |
| $CeNi_{0.9}Zr_{0.01}Y_{0.09}O_3$ | 42 L g$^{-1}$ h$^{-1}$ | 800 | 90 | 91 | [17] |
| $LaCr_{0.95}Ir_{0.05}O_{3-\delta}$ | 4000 h$^{-1}$ | 750 | 81 | 82 | [23] |
| 10 wt% Pd–$LaCr_{0.9}Ni_{0.1}O_{3-\delta}$ | 19.2 L g$^{-1}$ h$^{-1}$ | 750 | 63 | 96 | [47] |
| $LaNi_{0.05}Co_{0.05}Cr_{0.9}O_3$ | 12 L g$^{-1}$ h$^{-1}$ | 750 | 85 | 88 | [29] |
| $LaNi_{0.1}Cr_{0.9}O_3$ | 12 L g$^{-1}$ h$^{-1}$ | 750 | 84 | 87 | this work |

## 3. Materials and Methods

### 3.1. Catalyst Preparation

$LaNi_xCr_{1-x}O_3$ (x = 0.05, 0.1, 0.2, 0.3, 0.4, 0.5) catalyst precursors were synthesized using the sol-gel self-combustion method [25,48]. All the chemicals of analytical grade were purchased from Sinopharm Chemical Agent Company (Shanghai, China), including lanthanum oxide ($La_2O_3$), nitric acid, nickel nitrate hexahydrate ($Ni(NO_3)_2 \cdot 6H_2O$), chromium nitrate nonahydrate ($Cr(NO_3)_3 \cdot 9H_2O$), citric acid monohydrate ($C_6H_8O_7 \cdot H_2O$), and ammonia solution. $La_2O_3$ was entirely dissolved in nitric acid aqueous solution. A stoichiometric ratio of $Ni(NO_3)_2 \cdot 6H_2O$ and $Cr(NO_3)_3 \cdot 9H_2O$ was added to the solution under constant stirring. The mixed nitrate solution is combined with the complexing agent $C_6H_8O_7 \cdot H_2O$ (the ratio of metal ions and citric acid is 1.5:1). Ammonia solution (25% $NH_3$ by weight in water) was added to adjust the pH value of the solution to 7~9. After constant stirring at room temperature for a proper time, the mixed solution was heated on a heating platform until ignition. The flame temperature detected by an infrared detector was well above 1000 °C. Then, the product powder was collected and calcined at 700 °C in air for 4 h to remove residual organic chemicals. The obtained catalysts were in the spongy powder form, and their colors darkened with increasing x. The actual chemical composition of the catalysts determined by ICP is listed in Table 5. It is seen that the La content in the catalysts

is a little higher than the nominal composition while the Ni/Cr atomic ratios are close to the nominal ones.

**Table 5.** Chemical composition of the fresh catalysts determined by ICP.

| x in LaNi$_x$Cr$_{1-x}$O$_3$ | La:Ni:Cr (Atomic Ratio) |
|---|---|
| 0.05 | 100:4.5:91.0 |
| 0.1 | 100:9.1:86.3 |
| 0.2 | 100:19.0:76.5 |
| 0.3 | 100:27.9:66.7 |
| 0.4 | 100:38.0:55.6 |
| 0.5 | 100:47.3:48.6 |

### 3.2. Characterization

XRD. The crystalline phase structure of the catalyst samples was examined by an X-ray diffractometer (XRD, MXPAHF, MacScience, Kanagawa, Japan) using Cu K$\alpha$ radiation ($\lambda$ = 1.5406 Å) over the range of 2$\theta$ = 20–80° at room temperature.

XPS. X-ray photoelectron spectroscopy (XPS) analysis was performed using an electron spectrometer (ESCALAB 250, Thermo-VG Scientific, Waltham, MA, USA) with an exciting source of Al K$\alpha$ = 1486.6 eV.

TEM and EDS Mapping. The microstructures of the samples were observed by high-resolution transmission electron microscopy (HR-TEM, Talos F200X, FEI, Portland, OR, USA) and high-angle annular dark-field scanning transmission electron microscopy (HAADF-STEM, JEM-ARM200F, JEOL, Tokyo, Japan) operating at an accelerating voltage of 200 kV. The element distribution was measured by energy-dispersive X-ray spectroscopy mapping analysis (EDS-Mapping, Talos F200X, FEI, Portland, OR, USA)

TPR. Temperature-programmed reduction (TPR) was carried out with a simultaneous thermal analyzer (STA449F3, NETZSCH, Selb, Germany). A 10–15 mg powder sample was placed in an alumina crucible and degassed at 230 °C for 1 h to remove adsorbates. After cooling to room temperature, the sample was heated in situ in the flow of forming gas (5 vol% H$_2$/N$_2$, flow rate = 60 sccm) to 1000 °C with a heating rate of 10 °C min$^{-1}$. We take the first-order derivative on the thermogravimetric curve (TG) as DTG.

TPO. Temperature-programmed oxidation (TPO) was performed on the used catalysts to analyze the carbon deposition. The analysis was carried out with a simultaneous thermal analyzer. A 10–15 mg powder sample was placed in an alumina crucible. The sample was first heated under 10 sccm N$_2$ protection to 800 °C with a heating rate of 10 °C min$^{-1}$ to remove adsorbed gas molecules and to decompose the possible La$_2$O$_2$(CO$_3$). After cooling to room temperature, the sample was heated to 1000 °C in dry air (flow rate = 60 sccm) with a heating rate of 10 °C min$^{-1}$. The weight loss detected in the high-temperature stage above 500 °C reflects the amount of deposited carbon.

Specific surface area analysis. The BET-specific surface areas were measured by nitrogen adsorption at liquid nitrogen temperature (77 K) using a surface area analyzer (NOVA 3200e, Quantachrome, Boynton Beach, FL, USA). Before N$_2$ adsorption, the samples were degassed at 300 °C for 3 h to remove any residual moisture and other volatiles.

ICP-AES. The atomic ratios of La, Ni, and Cr in the fresh LaNi$_x$Cr$_{1-x}$O$_3$ samples were measured by inductively coupled plasma atomic emission spectrometry (ICP-AES) (Optima 7300 DV, PerkinElmer, Waltham, MA, USA). A 25 mg powder sample was dissolved in nitric acid aqueous solution under heated conditions. The obtained solution was diluted to ppm levels of metal ions to be measured by ICP-AES.

### 3.3. Catalytic Activity Tests

A 300 mg sample was placed in a fixed bed quartz reactor (i.d. = 6 mm) without dilution. The sample was heated to 700 °C in N$_2$ (30 sccm) and activated in pure H$_2$ (30 sccm) at 700 °C for 1 h before the DRM tests unless otherwise specified. After purging with N$_2$ for 30 min, the reactor was heated to the test temperature to carry out the catalyst

activity test under a continuous feed of approximately equimolecular $CO_2/CH_4$ mixture with a flow rate of 60 sccm without dilution. The same gaseous hourly space velocity (GHSV) of $1.2 \times 10^4$ mL $g_{cat}^{-1}$ $h^{-1}$ was maintained throughout the test. The steady-state tests were performed under atmospheric pressure at 750 °C. The reaction products were analyzed by on-line gas chromatography (GC9790, FULI, Taizhou, China), and the flow rate of the tail gas was measured by a soap film flowmeter. The conversions of $CH_4$ and $CO_2$ and the $H_2/CO$ ratio of $H_2$ and CO are defined as:

$$\text{Conv } CH_4 = \frac{[CH_4]_{in} - [CH_4]_{out}}{[CH_4]_{in}} \times 100\% \tag{1}$$

$$\text{Conv } CO_2 = \frac{[CO_2]_{in} - [CO_2]_{out}}{[CO_2]_{in}} \times 100\% \tag{2}$$

$$H_2/CO \text{ ratio} = \frac{[H_2]_{out}}{[CO]_{out}} \tag{3}$$

$$H_2 \text{ selectivity} = \frac{2 \times [H_2]_{out}}{[CH_4]_{in} - [CH_4]_{out}} \tag{4}$$

$$\text{Carbon balance} = \frac{[CH_4]_{out} + [CO_2]_{out} + [CO]_{out}}{[CH_4]_{in} + [CO_2]_{in}} \tag{5}$$

where $[CH_4]_{in}$ and $[CO_2]_{in}$ are the molar flow rates of the introduced $CH_4$ and $CO_2$, and $[CH_4]_{out}$, $[CO_2]_{out}$, $[H_2]_{out}$ and $[CO]_{out}$ are the molar flow rates of $CH_4$, $CO_2$, $H_2$ and CO in the tail gas.

## 4. Conclusions

$LaNi_xCr_{1-x}O_3$ (x = 0.05–0.5) samples in powder form were synthesized by the sol-gel self-combustion combustion method. Ni atoms are successfully doped into the $LaCrO_3$ perovskite lattice. The perovskite grains are polycrystalline, and the crystallite size decreases with increasing Ni content. The $CH_4$ conversion increases from 83% to 87% at 750 °C as the Ni loading increases from x = 0.05 to x = 0.5, meanwhile the carbon deposition rate increases from 0.02 to 76.2 mgc $g_{cat}^{-1}$ $h^{-1}$. The $CH_4$ and $CO_2$ conversions over the optimized sample (x = 0.1) are 83.9% and 87.1%, respectively, and the carbon deposition is negligible. We demonstrated that the $LaNi_xCr_{1-x}O_3$ perovskites show good stability and intrinsic catalytic activity for DRM reactions. We proposed that Ni atoms embedded on the surface of perovskite oxides (perovskite $LaNi_xCr_{1-x}O_3$ form or $LaNiO_\Delta$ submonolayer), are highly active owing to the open Ni 3d orbitals that resulted from oxygen vacancies. Such Ni atoms are atomically dispersed and act as the active centers for a DRM reaction. However, metal Ni nanoparticles usually coexist with the atomically dispersed Ni atoms embedded in the oxides, especially in samples with high Ni contents, which should be the main reason for carbon deposition. How to make the Ni-embedded perovskite catalysts more stable and suppress the formation of Ni nanoparticles still leaves an open question.

**Supplementary Materials:** The following supporting information can be downloaded at: https://www.mdpi.com/article/10.3390/catal12101143/s1, Figure S1: TEM images and EDS-Mapping of fresh $LaNi_{0.1}Cr_{0.9}O_3$ sample. (a) high magnification images to show the lattice structure and element distribution. The selected area (marked by a green square) is also magnified to reveal details of the lattice. (b) low magnification images to show the element distribution in a larger area; Figure S2: TEM images and EDS-Mapping of used $LaNi_{0.1}Cr_{0.9}O_3$ sample. (a) high magnification images to show the lattice structure and element distribution. The selected area (marked by a green square) is also magnified to reveal details of the lattice. (b) low magnification images to show the element distribution in a larger area; Figure S3: TEM images and EDS-Mapping of fresh $LaNi_{0.3}Cr_{0.7}O_3$ sample. (a) high magnification images to show the lattice structure and element distribution. The selected area (marked by a green square) is also magnified to reveal details of the lattice. (b) low magnification images to show the element distribution in a larger area; Figure S4: TEM images and

EDS-Mapping of used $LaNi_{0.3}Cr_{0.7}O_3$ sample. (a) high magnification images to show the lattice structure and element distribution. The selected area (marked by a green square) is also magnified to reveal details of the lattice. (b) low magnification images to show the element distribution in a larger area; Figure S5: TEM images and EDS-Mapping of fresh $LaNi_{0.5}Cr_{0.5}O_3$ sample; Figure S6: TEM images and EDS-Mapping of fresh $LaNi_{0.5}Cr_{0.5}O_3$ sample. (a) high magnification images to show the lattice structure and element distribution. The selected area (marked by a green square) is also magnified to reveal details of the lattice. (b) low magnification images to show the element distribution in a larger area; Figure S7: TEM images and EDS-Mapping of used $LaNi_{0.5}Cr_{0.5}O_3$ sample. (a) high magnification images to show the lattice structure and element distribution. The selected area (marked by a green square) is also magnified to reveal details of the lattice. (b) low magnification images to show the element distribution in a larger area; Figure S8: TEM images and EDS-Mapping of used $LaNi_{0.5}Cr_{0.5}O_3$ sample. The selected area (marked by green square) is also magnified to reveal details of the lattice; Table S1: Area ratio of $O_{ads}/O_{lat}$ for fresh and used $LaNi_xCr_{1-x}O_3$ samples in O 1s spectra.

**Author Contributions:** Conceptualization, T.Z.; data curation, T.Z.; formal analysis, F.Y.; investigation, H.Y. and X.T.; software, H.Y.; supervision, H.W.; writing—original draft, T.Z.; writing—review and editing, M.L. and H.W. All authors have read and agreed to the published version of the manuscript.

**Funding:** This research was funded by the National Natural Science Foundation of China (Grant no.: 21872129).

**Data Availability Statement:** All data included in this study are available upon request by contact with the corresponding author.

**Acknowledgments:** The authors thank the Instruments Center for Physical Science of the University of Science and Technology of China for the sample characterizations.

**Conflicts of Interest:** The authors declare no conflict of interest.

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
