# Peer review of "Influences of Ni Content on the Microstructural and Catalytic Properties of Perovskite LaNixCr1−xO3 for Dry Reforming of Methane"

_catalysts, doi:10.3390/catal12101143_

Round 1
Reviewer 1 Report
Manuscript ID: catalysts-1913750
Title: Influences of Ni Content on the Microstructural and Catalytic Properties of Perovskite LaNixCr1-xO3
In my opinion, the manuscript can be considered for publication after a major revision. The modifications suggested are below:
Ø Elemental analysis should be presented (ICP) for verification of desired composition.
Ø The calcination used is less than the reaction temperature, what effects this can cause. discuss it little bit deeply.
Ø In the catalytic performance, the hydrogen yield or selectivity should be performed.
Ø The carbon balance data relating to the catalytic performance measurements should be added
Ø Explain and prove reasons for having equal CH4 and CO2 conversions.
Ø In Table: 2. BET values are fluctuating, could you provide plausible reasons for that, and can you also state the % error of the quoted values?
Ø In figure 5, there is confusion about combining the TPR and TG, authors are asked to separate them in plotting and as well in the discussion.
Ø In Figure 7. DRM catalytic performance of the catalyst was displayed including the H2/CO. In the figure, the scale numbers of H2/CO should be added.
Ø References of the article must be upgraded. Suggested references include:
Ø Lanthanum–Cerium‐Modified Nickel Catalysts for Dry Reforming of Methane; https://doi.org/10.3390/catal12070715
Ø Role of Ca, Cr, Ga and Gd promotor over lanthana-zirconia– supported Ni catalyst towards H2-rich syngas production through dry reforming of methane, DOI: 10.1002/ese3.1063
Ø Modification of CeNi0.9Zr0.1O3 Perovskite Catalyst by Partially Substituting Yttrium with Zirconia in Dry Reforming of Methane; https://doi.org/10.3390/ma15103564
Ø In the reference list, the style of references should be adjusted in accordance with the journal format.
Reviewer 2 Report
The current manuscript deals with the analysis of the catalytic activity for DRM reactions of LaNixCr1-xO3 (x = 0.05 - 0.5) samples in powder form which were synthesized by the sol-gel self-combustion method.
The manuscript contains valuable information of interest for the wide scientific community interested in this kind of compounds. In my opinion the manuscript could be considered for publication with some observations:
1. The quality of the Figures could be improved for better understanding. For example, the Figure 6 a-d.
2. My recommendation is to add data from literature for comparison of catalytic performance for DRM reaction performance of LaNixCr1-xO3 with similar or different catalysts.
3. The conclusions could be a little expanded for the sake of clarity.
In summary, I will recommend this paper for publication after corrections.
Reviewer 3 Report
In the present work, the authors studied “Influences of Ni Content on the Microstructural and Catalytic Properties of Perovskite LaNixCr1-xO3”. The manuscript needs to be improved in the interpretation of results. As a reviewer, I would suggest that this manuscript needs to have major changes.
First of all, title the manuscript title needs to change, it is not suitable.
The abstract should be revised by using more scientific terms than using “native” kind of general terms and also repetitions such as the “self-combustion combustion method”.
In the introduction, a whole paragraph about SAC materials is not necessary as the authors are not working with the SAC so authors should refer to the previous literature, where B-site modification of LaBB’O3 was reported as a catalyst for reforming reactions (10.1039/c4ra07098d ;;; https://doi.org/10.1016/j.apcata.2018.07.039 ;;; https://doi.org/10.1016/j.apcatb.2017.10.022 ;;; http://dx.doi.org/10.1016/j.ijhydene.2015.12.075).
In XRD, theta values and planes should be reported.
What is the importance of the 022 planes and why authors want to show a shift with this plane? Even though it is not a highly intense peak. Why not present the high intense peak?
I will suggest authors also calculate lattice parameter values to discuss doping see literature (http://dx.doi.org/10.1016/j.ijhydene.2017.08.180 ;;;; 10.1007/s12039-017-1359-2).
Which plane is used to calculate the average grain size of perovskite crystals?
Elaborate on the discussion on the decrease in grain size with references.
The average crystal size of grains should also be present for reduced catalysts.
In TEM images, the d-spacing values of which elements or metal oxides should be mentioned in the script.
TEM images showing agglomeration of Ni are observed in used catalysts, on the other hand, why the grain size of perovskite (table 1) is decreased?
The dry reforming reaction is the metal active reaction the dispersion of metal plays a critical role, I suggest authors report the dispersion of metal in fresh and used catalysts.
XPS results should be presented with deconvolution of all the peaks.
Ni with different oxidation states should be discussed with elaborated discussion as well as Cr 2p XP spectra also.
In fresh catalysts why Cr6+ is increased with an increase in Ni should be discussed. In used catalysts, why there is no Cr6+ observed should be discussed.
O1s should also be added and discussed in relation to the different oxidation states.
In Fig 6, the activity results are presented very partially, H2/CO ratio should be added with time on stream, and CO2 conversion and H2/CO ratio should be added with temperature.
Ni dispersion is very high in the case of x=0.05 but its activity drops after 2h of reaction time. Why?
TPO results should be discussed in the manuscript..
I believe the authors need one more chance to provide more characterization study and good interpretation of data.
Round 2
Reviewer 1 Report
The standard of the current manuscript is well enough for publication
Reviewer 3 Report
In the present work, the authors studied “Influences of Ni Content on the Microstructural and Catalytic Properties of Perovskite LaNixCr1-xO3 for Dry Reforming of Methane”. The revised version of the manuscript is greatly improved in many sections that are pointed out by reviewers. As a reviewer, I would suggest that this manuscript is acceptable in its present form.
Finally, I would like to suggest authors make thorough language corrections.